# Level of Agreement, Reliability, and Minimal Detectable Change of the Musclelab^TM^ Laser Speed Device on Force–Velocity–Power Sprint Profiles in Division II Collegiate Athletes

**DOI:** 10.3390/sports10040057

**Published:** 2022-04-08

**Authors:** Jamie J. Ghigiarelli, Keith J. Ferrara, Kevin M. Poblete, Carl F. Valle, Adam M. Gonzalez, Katie M. Sell

**Affiliations:** 1Department of Allied Health and Kinesiology, Hofstra University, Hempstead, NY 11787, USA; ketelbop2015@gmail.com (K.M.P.); adam.m.gonzalez@hofstra.edu (A.M.G.); katie.sell@hofstra.edu (K.M.S.); 2Department of Athletics, Adelphi University, Garden City, NY 11530, USA; kferrara@adelphi.edu; 3Vortex Innovations, Westborough, MA 01581, USA; carlfvalle@gmail.com

**Keywords:** technology, testing, measurement, running, performance change

## Abstract

This study examined the level of agreement (Pearson product-moment correlation [*r*_P_]), within- and between-day reliability (intraclass correlation coefficient [ICC]), and minimal detectable change of the Musclelab^TM^ Laser Speed (MLS) device on sprint time and force–velocity–power profiles in Division II Collegiate athletes. Twenty-two athletes (soccer = 17, basketball = 2, volleyball = 3; 20.1 ± 1.5 y; 1.71 ± 0.11 m; 70.7 ± 12.5 kg) performed three 30-m (m) sprints on two separate occasions (seven days apart). Six time splits (5, 10, 15, 20, 25, and 30 m), horizontal force (*HZT* F_0_; N∙kg^−1^), peak velocity (V_MAX_; m∙s^−1^), horizontal power (*HZT* P_0_; W∙kg^−1^), and force–velocity slope (S_FV_; N·s·m^−1^·kg^−1^) were measured. Sprint data for the MLS were compared to the previously validated MySprint (MySp) app to assess for level of agreement. The MLS reported good to excellent reliability for within- and between-day trials (ICC = 0.69–0.98, ICC = 0.77–0.98, respectively). Despite a low level of agreement with *HZT* F_0_ (*r*_P_ = 0.44), the MLS had moderate to excellent agreement across nine variables (*r*_p_ = 0.68–0.98). Bland–Altman plots displayed significant proportional bias for V_MAX_ (mean difference = 0.31 m∙s^−1^, MLS < MySp). Overall, the MLS is in agreement with the MySp app and is a reliable device for assessing sprint times, V_MAX_, *HZT* P_0_, and S_FV_. Proportional bias should be considered for V_MAX_ when comparing the MLS to the MySp app.

## 1. Introduction

One focus of sprint research is the interpretation of the force-velocity-power (FVP) profile for guiding individualized program development [1]. The common FVP profile consists of theoretical horizontal force output per body mass (*HZT* F_0_; N·kg^−1^), theoretical maximal running velocity (V_MAX_; m·s^−1^), maximal mechanical horizontal power output per body mass (*HZT* P_0_; W·kg^−1^), the slope of the force–velocity curve (S_FV_), and the ratio of decreasing force with increasing velocity (D_RF_). Customizing training to the athlete’s specific FVP profile may allow coaches to develop optimal individual training interventions [1,2,3]. The benefits of FVP profiling include addressing variability in playing level [4,5], sport position [6], and age [7] as well as the ability to take into account external variables, such as time of season [8]. Team sports, such as rugby [3,9], soccer [4,7], ice hockey [10], and netball [11], as well as individual athletic activities, such as ballet [12] and gymnastics [13], have utilized this approach.

An important concern with FVP profiling is the ability to obtain reliable data for within- and between-day testing. Inconsistencies have been reported in jumping [14] and short (<10 m) sprinting performance [15], which causes concern about the efficacy of FVP profiling [16]. Therefore, equally important to the FVP profile is the training approach and its accuracy of measurement. The technology for monitoring FVP continues to improve in terms of efficiency, allowing coaches to assess and evaluate their players more quickly and effectively [17]. Currently, coaches and sports scientists have access to a variety of kinematic monitoring devices for sprinting performance, including high-speed cameras [18,19], iPhone apps [20], radar guns (i.e., Stalker II) [15], tether motor devices (i.e., 1080 Sprint) [21], inertial measurement units (IMUs) [22], global positioning systems (i.e., STATSports VIPER) (GPS) [23,24], and laser displacement measuring devices [25,26,27,28,29,30,31]. Although high-speed cameras and radar technology have shown to be valid methodologies for sprinting analysis, these systems are typically expensive and time consuming, requiring further analysis beyond the raw data captured. Contemporary systems, such as laser displacement measuring devices, provide immediate feedback and are easier to use during regular training sessions [21].

Studies on laser displacement measuring devices report intraclass correlation coefficients (ICCs) that range from moderate to good (0.64–0.83) [25] and good to excellent (ICC > 0.77) [27]. Recently, a commercially available device from Musclelab, Musclelab Laser Speed (MLS), was developed as a feasible means of assessing the FVP profile during sprinting. Van den Tillaar et al. [32] measured the validity of the MLS, combined with an IMU (MLS + IMU), in comparison to the use of force plates on step length, step velocity, and step frequency in trained sprinters. The authors tested 14 trained sprinters in the 50-m sprint using the MLS to record continuous distance over time. Significant positive correlations between the MLS + IMU and force plates for step velocity (*r* = 0.69) and step frequency (*r* = 0.53) were reported. In addition to kinematic data, the MLS calculates several of the values for the FVP profile, which have not been studied. Therefore, the first objective of this study is to examine the level of agreement of the FVP profile with the MLS with a previously validated field device, the MySprint (MySp) app [20], on maximal 30-m sprint time in a sample of Division Collegiate II athletes. Equivalence testing [33], such as the comparison of the MLS and the MySp app, can provide coaches with the information needed to compare instruments. The rationale for comparing commercial devices across the same movement exercise is used for other field devices, such as linear transducers [34,35]. This comparison provides coaches and practitioners with a greater understanding of which devices are practical to guide programming and training [34]. The results from this study can be used to help monitor training and provide direct feedback to the athlete.

The second objective is to report the minimal detectable change (MDC) data for the MLS. Minimal detectable change data are critical for practitioners to confidently identify actual change instead of typical within- and between-day variability [36]. Few studies have reported MDC for the FVP profile [9,36,37,38]; however, established MDC data are lacking, particularly in the case of laser technology. Given that sport lasers report excellent validity and reliability, we hypothesize that the MLS device will have a good level of agreement with the validated MySp app [20] and a high degree of reliability in terms of FVP profiling.

## 2. Materials and Methods

### 2.1. Experimental Design

A repeated measures design was used to assess within- and between-session reliability for sprint times at six splits (5, 10, 15, 20, 25, 30 m) and the FVP profile, which consisted of *HZT* F_0_, V_MAX_, *HZT* P_0_ and S_FV_. The S_FV_ for both devices was calculated using a published spreadsheet by Morin and Samozino [39]. The dependent variables were the six splits and the FVP profile. The independent variables were the number of sprint trials and testing days. The protocol consisted of each athlete performing three sprints on two occasions separated by seven days. To control for testing conditions, participants performed all sessions on the same day of the week, at the same time of day (~9 am), and on an indoor gymnasium floor to avoid the influence of weather. Participants were tested at the beginning of the week to ensure that they did not perform any team-organized strength training sessions 24 h prior to testing. Participants wore the same shoes to control for shoe–surface interface and were instructed not to perform any high-intensity training sessions 24 h prior to testing. All participants were compliant with the experimental guidelines.

### 2.2. Participants

Twenty-two Division II university athletes (20.1 ± 1.5 y; 1.71 ± 0.11 m; 70.7 ± 12.5 kg) volunteered to participate in this study. The sample included 16 females and 6 males (13 female soccer, 4 male soccer, 2 male basketball, 3 female volleyball). Based on an a priori power analysis using G*Power 3.1 software [40] for a one-tailed Pearson product-moment correlation between the two devices (MLS vs. MySp), we adopted an *α* = 0.05, ρ H_1_ correlation coefficient = 0.81, and ρ H_0_ correlation coefficient = 0.5. The sample size of 22 participants produced a power of 0.81. A post hoc power analysis was used to determine the actual power of the study as 0.95, using a sample size of 22 participants and a Pearson-product moment correlation as 0.87.

Inclusion criteria were (1) 18 to 24 years old, (2) active members of their respective sports teams, and (3) free of any physical limitations, defined as having no lower or upper body musculoskeletal injuries that affected maximum sprinting ability. Exclusion criterion included having a musculoskeletal upper or lower body injury that affected the sprinting exercise. Before enrollment, all participants completed the Physical Activity Readiness Questionnaire and a medical history questionnaire.

### 2.3. Procedures

Participants reported to the research facility on two days for sprint testing, and each visit was separated by seven days. Prior to maximal sprint testing, participants were weighed on a digital scale (Taylor Precision Products, Oak Brook, IL, USA) with full clothing and shoes, and their body mass in kilograms was entered into the MLS and MySp app. Next, participants performed a standardized 15 min warm-up, consisting of 5 min of jogging, 5 min of lower limb dynamic stretching, and 5 min of progressive 30-m sprints at 50%, 70% and 90% effort. After the warm-up, participants performed three 30-m sprints at maximal effort, with 5 min rest in between sprints. A 30-m sprint distance is used in previous research [15,27,36] and is the distance setting for the MySp app. All participants started in a two-point stance, with no false step, and were instructed to start the sprint at any time. All participants were familiar with sprint testing, and all warm-ups and sprinting sessions were supervised by a certified strength and conditioning specialist (CSCS-NSCA). Each sprint was simultaneously assessed by the MLS and MySp app technology.

### 2.4. Musclelab Laser Speed

Measurements of instantaneous split times and FVP data were recorded using MLS (Musclelab^TM^ 6000 ML6LDU02 Laser Speed device, Ergotest Innovations, Stathelle, Norway), sampling at 2.5 KHz (Figure 1). Raw data were analyzed using the Musclelab software (version 10.213.98.5188), which measures continuous velocity (V_h_ (t)) and distance to create an individual FVP profile. A mathematical model is fitted to the recorded velocity/time by calculating the time constant, “tau” (τ). Thus, the velocity/time can be retrieved from the model as
V_h_ (t) = V_MAX_ (1 − *e*^−t/^^τ^),(1)
where V_MAX_ is observed maximal velocity. A mathematical derivation allows calculation of horizontal acceleration/time:a_h_ (t) = V_MAX/_τ ∗ *e*^−t/^^τ^,(2)

The horizontal force/time can then be calculated:F_(h)_ (t) = ma_h_ + F_air_, (3)
where F_air_ is force caused by wind drag. F_h_ (t) is then expressed as a function of V_h_ (t), and a linear fit is applied. It gives the form F_h_ = Av_h_ + B, also known as F/V profile, where A and B are polynomial constants. The MLS is designed to operate in a typical indoor environment, such as a sports hall or gymnasium, on a flat surface of 29–60 m. MLS was placed 3 m behind the starting line on a tripod of a height of 0.91 m and oriented to the participant’s lower back proximal to his or her center of mass. The distance behind the start line for the MLS is similar to previous methodology [25]. The MLS has a standard aiming scope (Strike Red^®^ Dot 1 × 30) with a pointer beam (605 nm) precision width of 1–5 mm and a range of 75 m. The MLS measures continuous distance and FVP profile during sprinting. To ensure that all participants performed the sprint in a straight line, the investigators created a lane width of 0.66 m, which is considerably less than a standard track, ranging from 1.07–1.22 m, depending on the level of competition. Sprint performance measurements were available immediately after each sprint. The head strength and conditioning coach was the sole operator for all MLS sprint trials.

### 2.5. MySp App

The MySp app videos were filmed with an iPad (7th generation; iOS 14.4.2 with built-in slow-motion video support at 120 fps at a quality of 720p) according to previously validated methodologies [20]. Notably, Romero et al. [20] used a 40-m track, whereas the current study implemented a 30-m track, consequently slightly altering our parallax measurements (Figure 2). The iPad was mounted to a tripod (height, 1.46 m) to record each sprint, assessing the frontal plane to film the sprint from the side. The video parallax was corrected to ensure that the 5, 10, 15, 20, 25, and 30 m split times were measured accurately. As per [20], the marking poles were not exactly at the associated distances but, rather, at the adjusted positions. The iPad camera filmed the participants’ hips as they crossed the markers when they were precisely at the targeted distances (Figure 3).

### 2.6. Independent Observers

Two independent observers were asked to time stamp the exact start of the sprint initiated by the participants’ movement and each split time for the entire sprint. The rationale for two independent observers using the MySp app was to ensure that each observer accurately identified the start of the sprint and the crossing of the hips (i.e., split times) at each marking point. The start of the sprint was defined as the moment when the back leg plantar flexed, indicating that the participant applied force to the ground. Stamping the start time of the participants’ movement, instead of from an external verbal cue (e.g., “set, go”), avoided any confounding effects of participants’ reaction time. The first three successful repetitions for each testing day were used in the analysis.

### 2.7. Statistical Analysis

Data analysis was performed using IBM SPSS, Version 27.0 (SPSS Inc., Chicago, IL, USA) software for Windows. Descriptive data for participants’ characteristics and experimental variables were calculated as means and standard deviations with 95% confidence intervals. Normality of the distributions for each dependent variable was tested using a Shapiro–Wilk test. Absolute reliability was calculated using ICCs for the within-day (within the three sprints of each session) and between-day (mean of each session) data to determine the reliability of the MLS.

The ICC results were interpreted as 0.2–0.49 = low, 0.50–0.74 = moderate, 0.75–0.89 = high, 0.9–0.98 = very high, and 0.99 = extremely high [41]. Internal consistency for the MLS was examined by calculating the coefficient of variation (CV). A coefficient of variation (SD/mean) of <10% was considered acceptable for reliability and <5%, acceptable for fitness testing standards [42]. The CV was calculated for each participant for the three sprints across both testing days. The average CV of all participants for each dependent variable was reported.

The validated MySp app was selected as the device to assess the level of agreement with the MLS [20]. Level of agreement was assessed by an examination of the significance of the paired *t*-test mean difference, ICCs, and Pearson product-moment correlation (*r*_P_) with the 95% confidence intervals. Bland–Altman plots were created (GraphPad Prism version 9.2 for Windows; GraphPad software, La Jolla, CA, USA) to examine the linear regression analysis of the difference and average scores across the devices [20,43]. Examination of the 95% confidence intervals for the slope and *y*-intercept across the split times and FVP were used to determine proportional and fixed bias, respectively.

A one-way repeated-measures analysis of variance (ANOVA) was used to calculate the mean square error for the within- and between-session data. The standard error of measurement (SEM) was calculated by taking the square root of the mean square error [44], and the MDC was detected at the 95% confidence interval (MDC_95_). The MDC_95_ provided assurance that a true change had occurred, outside of error; MDC_95_ = SEM × √2 × 1.96 [45]. The smallest worthwhile change (SWC) was calculated for within-day testing by selecting the best sprint trial for each session, then multiplying the between-participant standard deviation by 0.2. For between-day testing, the average standard deviation across both days was multiplied by 0.2 [46].

To ensure the accuracy of the MySp app analysis, two independent observers analyzed each participant’s video for sprint times. Six independent *t*-tests were used to calculate the mean differences across the six splits (5, 10, 15, 20, 25, 30 m) to determine whether significant differences in sprint times existed between observers. If no significant differences were found, the data from the primary investigator was used in the analysis. If significant differences were found, the average of the two observers was recorded.

## 3. Results

### 3.1. Within-Day Reliability

Normality was satisfied for 95.8% of the data, using a Shapiro–Wilks test (*p* > 0.05). Out of 120 variables (10 scores x 6 sprints x 2 devices), five variables did not satisfy normality; (1) MLS *HZT* F_0_ for Day 1, Sprint 2; (2) MLS *HZT* F_0_ for Day 1, Sprint 3; (3) MySp app *HZT* P_0_ for Day 2, Sprint 3; (4) MySp app 10-m time for Day 2, Sprint 3; and (5) MLS S_FV_ for Day 1, Sprint 1.

Raw data means, standard deviations, ICCs, CVs, SWCs, SEMs, and MDC_95_ values with 95% confidence intervals were calculated for the sprint times and FVP profile for the three sprints across both testing days (Table 1, Table 2, Table 3 and Table 4). Except for *HZT* F_0_ (0.71 and 0.69) and Day 1 for S_FV_ (0.71); nine of the ten variables reported high to very high (0.8–0.98) within-day reliability scores for both days. The average ICCs across all variables for Day 1 and Day 2 were 0.90 and 0.92, respectively. The average coefficients of variation across all variables for Day 1 and Day 2 were highly acceptable, at 2.4% and 1.8%, respectively. The MDC_95_ values for split times ranged from 0.06 to 0.11 s on Day 1 and 0.05 to 0.14 s on Day 2. The MDC_95_ values for FVP on Day 1 were 1.16 N·kg^−1^ (*HZT* F_0_), 0.25 m·s^−1^ (V_MAX_), 1.99 W·kg^−1^ (*HZT* P_0_) and 0.29 N·s·m^−1^·kg^−1^ (S_FV_); and for Day 2, were 0.83 N·kg^−1^ (*HZT* F_0_), 0.30 m·s^−1^ (V_MAX_), 1.41 W·kg^−1^ (*HZT* P_0_), and 0.12 N·s·m^−1^·kg^−1^ (S_FV_).

### 3.2. Between-Day Reliability

Between-day reliability values are presented in Table 5 as an average across both testing days. Seven out of ten scores were above 0.9 in addition to all CVs ≤5%, indicating very good to acceptable reliability. The average ICC and CV were 0.90 and 1.6%, respectively, across all variables between testing days. For the time between the days, the range of MDC_95_ values for split times was 0.08–0.14, and values for FVP were 0.66 N·kg^−1^ (*HZT* F_0_), 0.28 m·s^−1^ (V_MAX_), 1.43 W·kg^−1^ (*HZT* P_0_), and 0.11 N·s·m^−1^·kg^−1^ (S_FV_).

### 3.3. MySp App Reliability

The descriptive and reliability data for the MySp app are reported in Table 6. The average ICCs across all variables for Day 1 and Day 2 were excellent, at 0.91 and 0.90, respectively. The average coefficients of variation across all variables for Day 1 and Day 2 were highly acceptable, at 2.6% and 2.5%, respectively.

### 3.4. Level of Agreement

The mean difference score and *r*_P_ were calculated to assess the level of agreement between MLS and MySp app (Table 7). The *r*_P_ ranged from 0.44 to 0.98, with low agreement for *HZT* F_0_ (0.44), moderate for S_FV_ (0.60), and the remaining variables showing high levels of agreement, greater than 0.88. Three of the ten variables had no statistical difference in mean difference scores (*p* > 0.05) based on the paired *t*-test. A statistical difference (*p* < 0.05) was reported for four split times 15–30 m with the MySp reporting faster times than the MLS. A statistical difference (*p* < 0.05) was reported for *HZT* F_0_ with the MySp app reporting lower *HZT* F_0_ values compared to MLS (6.85 < 7.1, mean difference = 0.25). A statistical difference (*p* < 0.001) was reported for V_MAX_ (MySp faster than MLS, 8.08 > 7.77 m·s^−1^, mean difference = 0.31 m·s^−1^) and S_FV_ (*p* = 0.01), MySp > MLS, −0.83 > −0.87, mean difference = 0.04. No proportional or fixed bias was observed for split times, *HZT* F_0_, *HZT* P_0_, and S_FV_, indicated visually by the Bland–Altman plots (Figure 4, Figure 5, Figure 6 and Figure 7), which show a low *R*^2^ for the linear regression of *HZT* F_0_, *HZT* P_0_, and S_FV_ (0.12, 0.14 and 0.11, respectively). In addition, except for V_MAX_, all confidence intervals, including the split times, contained zero for the slope and intercept, indicating that the differences between the devices were the same across the six sprints. For V_MAX_, the *R*^2^ of the linear regression plots has a large effect size, 0.24. The 95% confidence interval for the slope did not contain zero CI (−0.2–−0.02), thus showing proportional bias toward the MySp app. The negative slope (−0.11) indicated that the MySp app had a greater proportional bias for the faster runners.

### 3.5. Inter-Observer Analysis

Six independent *t*-tests were used to determine significant differences among six sprint times between two independent observers. Mean difference and *p*-values for the 5 m (0.001 ± 0.01 s, *p* = 0.932), 10 m (0.008 ± 0.02 s, *p* = 0.735), 15 m (0.01 ± 0.03 s, *p* = 0.731), 20 m (0.01 ± 0.04 s, *p* = 0.756), 25 m (0.02 ± 0.05 s, *p* = 0.774), and 30 m (0.02 ± 0.07 s, *p* = 0.794) revealed no significant differences for sprint times between the observers.

## 4. Discussion

This study reported the level of agreement, reliability, and MDC of the MLS device during maximal 30-m sprints in a sample of Division II Collegiate athletes. The major finding is that the MLS displayed a moderate to excellent level of agreement and reliability for nine of ten variables, with the exception of *HZT* F_0_. A second finding is that there was a significant proportional bias for V_MAX_ as compared the MLS to MySp app, with the MySp app detecting faster velocities (mean difference = 0.31 m·s^−1^). Finally, significant differences occurred in four split times (15–30 m), *HZT* F_0_, and S_FV_ between the two devices, with the MySp app having faster times, lower *HZT* F_0_, and a velocity-dominant slope.

The low to moderate ICC and *r*_P_ for *HZT* F_0_ and moderate ICC and *r*_P_ for S_FV_ indicate a partial rejection of our primary hypothesis that the MLS would agree with the MySp app for all components of the FVP profile. Our second hypothesis that the MLS is reliable for within- and between-day testing for split times and the FVP profile was primarily supported for nine of ten variables. Despite proportional bias for V_MAX_, our finding is consistent with a previous study showing the mathematical model for the MySp app to have a 0.32 m·s^−1^ bias compared to force plate analysis [47].

Our ICCs were both higher [25,28] and similar [27,31] as compared to previous studies on the reliability of sports lasers. Within- and between-day ICCs and CVs across all split times were above 0.83 and less than 5%, respectively. Both *HZT* P_0_ and V_MAX_ within- and between-day data showed ICCs and CVs above 0.92 and less than 5%. Notably, *HZT* F_0_ showed acceptable internal consistency with CVs less than 5% for both within- (4.7% and 3.8%) and between-day (2.7%) sessions. Pearson-product moment correlations and absolute agreement ICCs for all split times, *HZT* P_0_, V_MAX,_ and S_FV_ ranged from 0.60 to 0.98 and 0.68 to 0.95. Similar to our reliability data, the MLS did not agree with the MySp app data for *HZT* F_0_, with *r*_P_ and ICC at 0.44 and 0.55, respectively. Bland–Altman plots show no fixed or proportional bias, except for proportional bias for V_MAX_ (MySp > MLS). The absence of bias is supported by all split times, *HZT* F_0_, *HZT* P_0_, and S_FV_ as having zero in the 95% CI for the slope and intercept. V_MAX_ did not have zero in the 95% CI for slope (−0.2–−0.02) but did have zero for the intercept (−0.15–1.29), indicating no fixed, but proportional, bias.

MDC is defined as the smallest change in a variable that reflects a true change in performance [48]. MDC is important, particularly when monitoring athletes over several trials, as between-trial variation may suggest a change that has not exceeded a threshold error [36,49]. Studies by Edwards et al. [36] and Ferro et al. [37] used radar and laser technology, respectively, reported MDCs at the 90% confidence interval. Our split time and FVP profile MDC data were similar to those of Edwards et al. (2021) when comparing the average of three trials. SWC is defined as the smallest change in a metric that is likely of practical importance [50]. In four splits (15–30 m) and V_MAX_, the CV% was approximately equal to the SWC%, indicating that the MLS had an “okay” sensitivity rating in terms of detecting real change [51]. In contrast, the CV% for *HZT* F_0_, *HZT* P_0_, and S_FV_ was greater than the SWC%. For Day 1 and Day 2, *HZT* F_0_ was CV (4.7%) > SWC (1.6%) and 3.8% > 1.2%, *HZT* P_0_ was 3.7% > 2.5% and 3.5% > 2.2%, and S_FV_ was 7.0% > 1.5% and 4.0% > 1.1%. Therefore, all three measurements had “marginal to poor” within-day sensitivity [51,52].

Coaches find FVP profiling to be useful as they allow for a more individualized training approach, and if the correct data collection methodology is used, FVP can provide accurate monitoring of progression [1]. However, the value of FVP profiling is subject to debate [14,16]. Studies report *HZT* F_0_ to have moderate reliability [15,17,36,53], specifically for sprints < 10 m [15,36]. Our data are consistent with this finding for *HZT* F_0_, which achieved moderate ICCs for within- (0.71 and 0.69) and between-day (0.77) testing. Similar inconsistencies for *HZT* F_0_ have been reported when monitoring jumping with large variations between trials noted [14].

Comparisons between the MLS and MySp app can be made in terms of their practical uses in real-world settings considering that both are accurate and valid sprint testing devices. The first advantage of the MLS is that it provides immediate feedback to the coach and sprinter allowing for better coaching instruction during the training session, while the MySp app uses video technology requiring further analysis post-sprint. The second is that the MLS accommodates a variety of settings, such as a field or gymnasium, with minimal setup time, while the MySp app requires a 15- to 20-min setup time (with two people) to ensure that all of the distances and parallaxes are accurate and that the viewing area is clear. This kind of setup can be challenging for short-staffed strength and conditioning coaches who might have one coach per training group. A final advantage is that the MLS can be combined with other measurement devices (e.g., IMU, contact grid) to access more sophisticated data, such as step velocity, step length, and contact time [32]. However, the MLS costs approximately $6000, whereas the MySp app is $10. Further, the MLS does not display the entire FVP profile, as does the MySp app. In this regard, [27] suggests that S_FV_ and D_RF_ are the most important factors in the FVP profile, which the MLS does not record, but the MySp app does. Nevertheless, according to [27], the use of laser technology with trained sprinters resulted in poor reliability for both S_FV_ and D_RF_.

This study has certain limitations. A research-grade criterion variable, such as a high-speed camera or force plate system, which the study lacked, would provide a better means to validate the MLS device. The rationale for using the MySp app was that the investigators wanted to provide two easily accessible devices for practitioners to make the comparison. A suggestion for future research would be a follow-up validation study that compares the MLS to a criterion variable. Second, the significant difference across the four split times, *HZT* F_0_, and proportional bias for V_MAX_ between the MLS and MySp app could be due to the sampling rate of the iPad camera (i.e., 120 fps), which is equivalent to a high-speed smartphone [19]. Romeo et al. [20] suggest using an iPhone at a sampling rate of 240 fps when utilizing the MySp app. A camera with a higher sampling rate might have reduced some of the statistical differences across the split times and our bias due to providing a more accurate identification point of the sprinter’s start and the hips as they crossed the specified marking poles. Notably, previous literature has used 120 fps to assess sprint performance [54] and treadmill running [55]; however, a practical suggestion for coaches is to have an iPhone, as opposed to an iPad, readily available when using the MySp app. Third, the inherent limitations of testing a sample of novice sprinters may have caused variation in our *HZT* F_0_ data. A contributor to the unwanted variation in *HZT* F_0_, particularly in the early phases of the sprint (<10 m), is that the changes in the lumbar point (where the laser is aimed) to the center of mass of the sprinter decreases as the sprinter continues to rise to an upright posture [30]. This notion is supported by Talukdar et al. [56], who reported *HZT* F_0_ to have the highest CV in their data set of young female team sport athletes (nevertheless, this study reported acceptable overall reliability for *HZT* F_0_ [ICC = 0.89, CI 0.77–0.94], using radar). One reason for this variation may be that novice sprinters rise too early and are inconsistent at the start of the sprint, and, thus, are not consistent in applying force at the start [56]. The start positon (two-point vs. three-point) maybe a source of error for *HZT* F_0_ due to different athletes becoming accustomed to different starts. The current study used a two-point stance, which is similar to recent research when assessing 30 m sprint performance [57]. Nonetheless, teaching athletes to have more consistency in their start position and sprinting technique (i.e., avoid rising too fast) in the first 10 m may reduce this variation. Therefore, caution should be used with *HZT* F_0_ data when evaluating progression and the training prescription.

## 5. Conclusions

The current study found that the MLS displays excellent agreement with the MySp app for most performance measures and that the MLS is a reliable (within- and between-day) device for measuring 30-m split times, V_MAX_, *HZT* P_0_, and S_FV_. Nevertheless, the MLS has moderate to poor accuracy in measuring *HZT* F_0_. Coaches and practitioners need to be aware of the significant proportional bias for V_MAX_, with the MySp app’s reporting higher sprint velocities than the MLS in the faster runners and a velocity dominant slope for S_FV_. The MLS is sensitive to a change in 15–30 m sprints and V_MAX_. Finally, the MDC data add to the knowledge available for coaches and practitioners in terms of identifying an actual change in performance.

## Figures and Tables

**Figure 1 sports-10-00057-f001:**
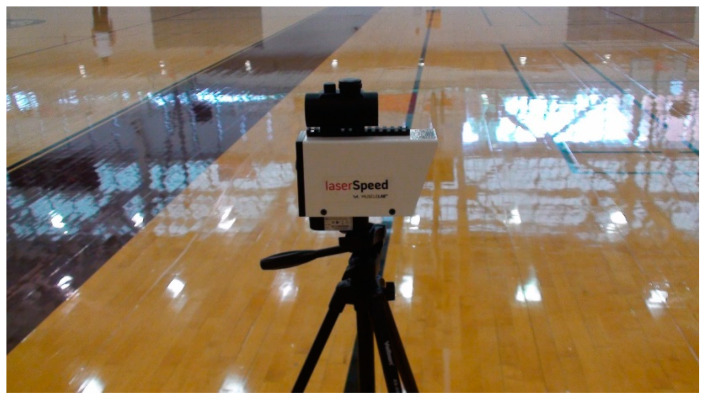
Musclelab^TM^ Laser Speed with tripod.

**Figure 2 sports-10-00057-f002:**
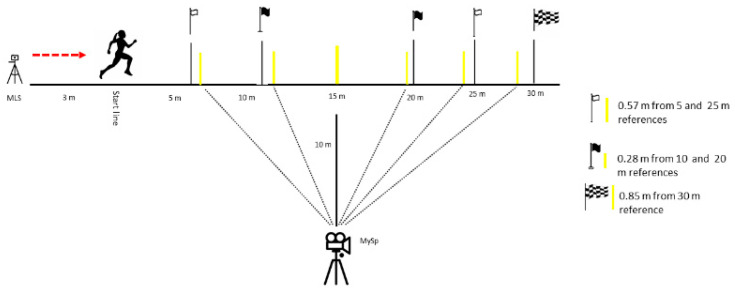
Experimental setup of the testing sessions for the validation of the Musclelab^TM^ Laser Speed and MySprint app. The yellow lines represent the marking poles as part of the, Mysprint set up.

**Figure 3 sports-10-00057-f003:**
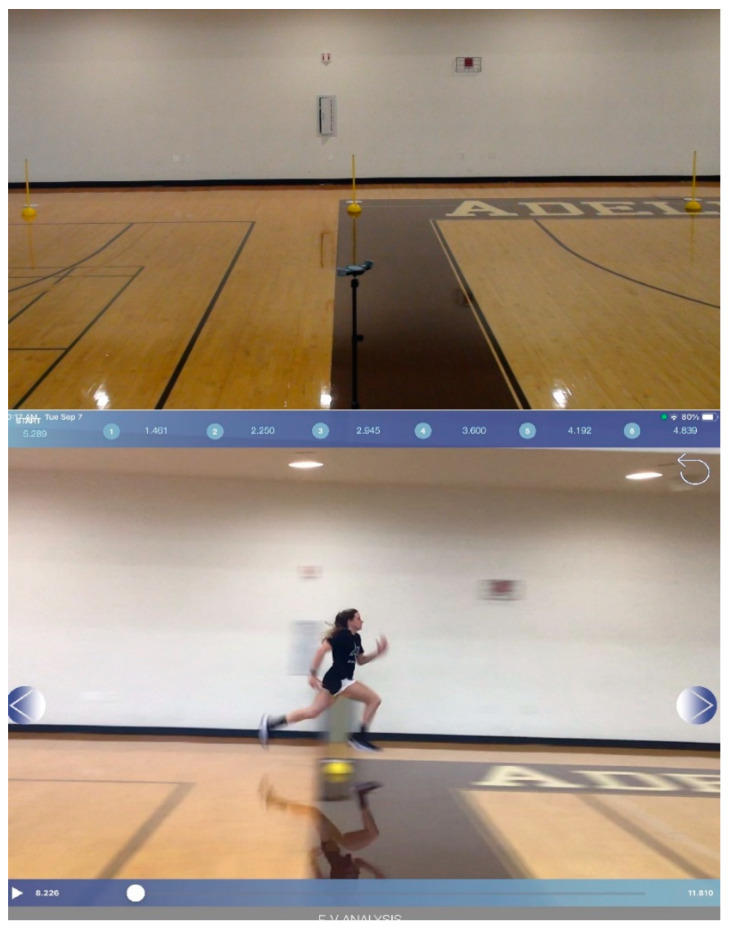
Pictorial representation of the MySprint app.

**Figure 4 sports-10-00057-f004:**
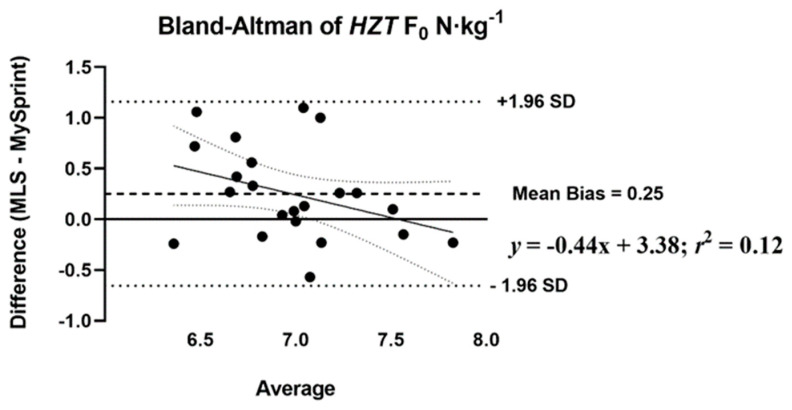
Bland-Altman plot that provides a comparison of the differences and the average between the Musclelab^TM^ Laser Speed and the MySprint app for *HZT* F_0_ N·kg^−1^. The upper and lower lines represent ± 1.96 N·kg^−1^.

**Figure 5 sports-10-00057-f005:**
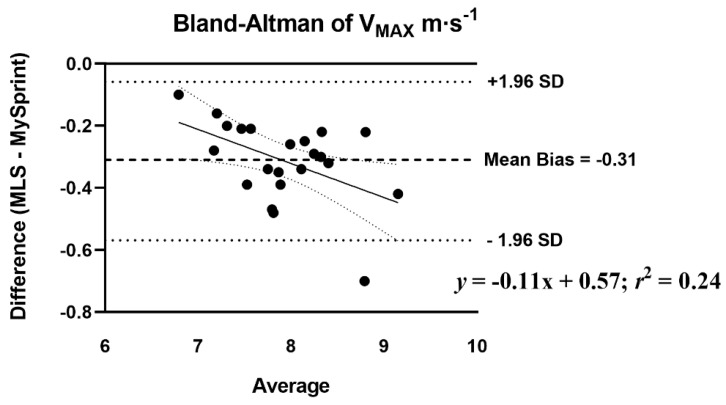
Bland–Altman plot that provides a comparison of differences and the average between the Musclelab^TM^ Laser Speed and the MySprint app for V_MAX_ m·s^−1^. The upper and lower lines represent ± 1.96 m·s^−1^.

**Figure 6 sports-10-00057-f006:**
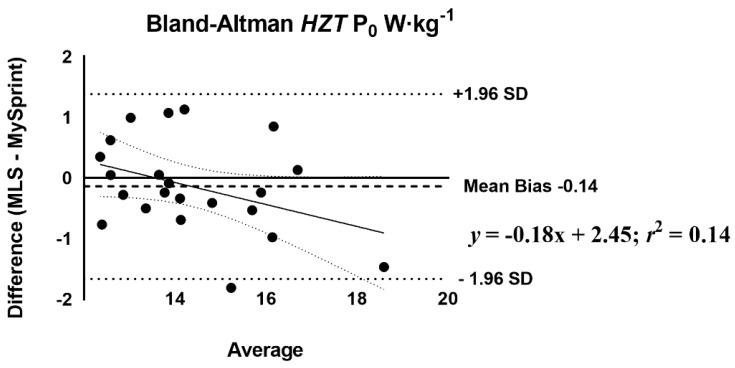
Bland–Altman plot that provides a comparison of the differences and the average between the Musclelab^TM^ Laser Speed and the MySprint app for *HZT* P_0_ W·kg^−1^. The upper and lower lines represent ± 1.96 W·kg^−1^.

**Figure 7 sports-10-00057-f007:**
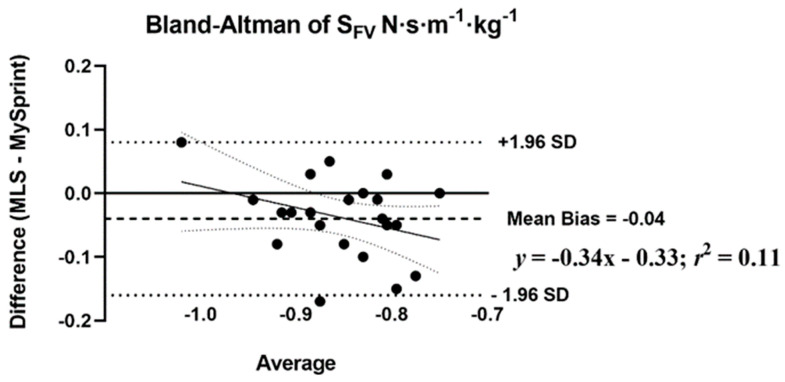
Bland–Altman plot that provides a comparison of the differences and the average between the Musclelab^TM^ Laser Speed and the MySprint app for S_FV_ N·s·m^−1^·kg^−1^. The upper and lower lines represent ± 1.96 N·s·m^−1^·kg^−1^.

**Table 1 sports-10-00057-t001:** Means ± standard deviation (95% confidence interval) across three sprints for Musclelab^TM^ Laser Speed on Day 1, within-day trials.

Performance	Sprint 1	Sprint 2	Sprint 3	Mean ± SD
5-m (s)	1.45 ± 0.05 (1.42–1.47)	1.44 ± 0.04 (1.43–1.46)	1.44 ± 0.04 (1.42–1.46)	1.45 ± 0.04 (1.43–1.47)
10-m (s)	2.24 ± 0.08 (2.20–2.28)	2.23 ± 0.07 (2.20–2.27)	2.23 ± 0.08 (2.19–2.27)	2.24 ± 0.07 (2.20–2.27)
15-m (s)	2.95 ± 0.11 (2.90–3.00)	2.94 ± 0.10 (2.90–2.99)	2.94 ± 0.12 (2.88–2.99)	2.95 ± 0.11 (2.90–3.00)
20-m (s)	3.63 ± 0.15 (3.56–3.70)	3.62 ± 0.14 (3.56–3.69)	3.62 ± 0.16 (3.54–3.62)	3.63 ± 0.14 (3.56–3.69)
25-m (s)	4.29 ± 0.18 (4.21–4.38)	4.28 ± 0.18 (4.20–4.37)	4.28 ± 0.20 (4.18–4.37)	4.29 ± 0.18 (4.21–4.37)
30-m (s)	4.95 ± 0.23 (4.84–5.06)	4.93 ± 0.22 (4.82–5.04)	4.93 ± 0.25 (4.81–5.05)	4.94 ± 0.22 (4.84–5.04)
*HZT* F_0_ N·kg^−1^	7.10 ± 0.55 (6.85–7.34)	7.11 ± 0.55 (6.86–7.35)	7.22 ± 0.62 (6.94–7.49)	7.14 ± 0.45 (6.94–7.34)
V_MAX_ m·s^−1^	7.78 ± 0.53 (7.54–8.02)	7.79 ± 0.51 (7.56–8.02)	7.80 ± 0.57 (7.55–8.06)	7.79 ± 0.53 (7.55–8.03)
*HZT* P_0_ W·kg^−1^	14.29 ± 1.54 (13.60–14.97)	14.31 ± 1.63 (13.59–15.04)	14.71 ± 1.84 (13.89–15.53)	14.44 ± 1.57 (13.74–15.13)
S_FV_ N·s·m^−1^·kg^−1^	−0.87 ± 0.07 (−0.90–−0.84)	−0.85 ± 0.11 (−0.90–−0.79)	−0.87 ± 0.07 (−0.90–−0.84)	−0.86 ± 0.06 (−0.89–−0.83)

**Table 2 sports-10-00057-t002:** Absolute reliability statistics for Musclelab^TM^ Laser Speed on Day 1; within-day trials.

Performance	ICC	CV%	SWC	SEM	MDC_95_
5-m (s)	0.87 (0.72–0.94)	2.0 (0.7–3.2)	0.01 (0.6%)	0.02	0.06
10-m (s)	0.93 (0.87–0.97)	1.5 (0.6–2.3)	0.01 (0.4%)	0.03	0.08
15-m (s)	0.96 (0.92–0.98)	1.2 (0.5–1.9)	0.02 (0.7%)	0.03	0.08
20-m (s)	0.97 (0.95–0.99)	1.1 (0.5–1.6)	0.03 (0.8%)	0.04	0.11
25-m (s)	0.98 (0.96–0.99)	0.9 (0.4–1.4)	0.04 (0.9%)	0.04	0.11
30-m (s)	0.98 (0.97–0.99)	0.9 (0.5–1.3)	0.04 (0.8%)	0.04	0.11
*HZT* F_0_ N·kg^−1^	0.71 (0.41–0.87)	4.7 (2.9–6.5)	0.12 (1.6%)	0.42	1.16
V_MAX_ m·s^−1^	0.99 (0.97–0.99)	1.0 (0.7–1.2)	0.11 (1.4%)	0.09	0.25
*HZT* P_0_ W·kg^−1^	0.92 (0.85–0.96)	3.7 (2.1–5.3)	0.37 (2.5%)	0.72	1.99
S_FV_ N·s·m^−1^·kg^−1^	0.71 (0.39–0.88)	7.0 (0.03–0.1)	0.01 (1.5%)	0.10	0.29

*Note.* ICC = intraclass correlation coefficient; CV% = coefficient of variation; SWC = smallest worthwhile change (left side), percentage of mean (right side); SEM = standard error of measurement; MDC_95_ = minimal detectable change at 95% confidence interval.

**Table 3 sports-10-00057-t003:** Means ± standard deviation (95% confidence interval) across three sprints for Musclelab^TM^ Laser Speed on Day 2; within-day trials.

Performance	Sprint 1	Sprint 2	Sprint 3	Mean ± SD
5-m (s)	1.46 ± 0.05 (1.43–1.48)	1.44 ± 0.04 (1.42–1.46)	1.44 ± 0.04 (1.42–1.46)	1.44 ± 0.04 (1.42–1.46)
10-m (s)	2.26 ± 0.09 (2.22–2.31)	2.23 ± 0.07 (2.20–2.27)	2.23 ± 0.08 (2.20–2.27)	2.24 ± 0.08 (2.21–2.28)
15-m (s)	2.99 ± 0.12 (2.93–3.04)	2.95 ± 0.11 (2.90–3.00)	2.95 ± 0.11 (2.96–3.01)	2.96 ± 0.11 (2.91–3.02)
20-m (s)	3.67 ± 0.17 (3.59–3.75)	3.63 ± 0.15 (3.56–3.69)	3.63 ± 0.15 (3.56–3.70)	3.64 ± 0.15 (3.57–3.71)
25-m (s)	4.34 ± 0.21 (4.24–4.44)	4.29 ± 0.18 (4.21–4.38)	4.30 ± 0.20 (4.21–4.39)	4.31 ± 0.19 (4.22–4.40)
30-m (s)	5.00 ± 0.25 (4.89–5.12)	4.95 ± 0.23 (4.85–5.05)	4.96 ± 0.24 (4.85–5.07)	4.97 ± 0.24 (4.86–5.08)
*HZT* F_0_ N·kg^−1^	6.93 ± 0.42 (6.74–7.12)	7.12 ± 0.43 (6.92–7.31)	7.12 ± 0.37 (6.95–7.29)	7.06 ± 0.32 (6.91–7.20)
V_MAX_ m·s^−1^	7.71 ± 0.58 (7.45–7.97)	7.76 ± 0.55 (7.51–8.00)	7.76 ± 0.61 (7.48–8.03)	7.74 ± 0.57 (7.49–8.00)
*HZT* P_0_ W·kg^−1^	13.83 ± 1.70 (13.08–14.59)	14.31 ± 1.59 (13.60–15.01)	14.27 ± 1.56 (13.57–14.96)	14.14 ± 1.56 (13.44–14.83)
S_FV_ N·s·m^−1^·kg^−1^	−0.85 ± 0.06 (−0.88–−0.83)	−0.88 ± 0.06 (−0.91–−0.85)	−0.89 ± 0.08 (−0.93–−0.85)	−0.87 ± 0.06 (−0.90–−0.85)

**Table 4 sports-10-00057-t004:** Absolute reliability statistics for Musclelab^TM^ Laser Speed on Day 2; within-day trials.

Performance	ICC	CV%	SWC	SEM	MDC_95_
5-m (s)	0.91 (0.81–0.96)	1.3 (0.9–1.7)	0.01 (0.6%)	0.02	0.05
10-m (s)	0.94 (0.87–0.97)	1.2 (0.8–1.5)	0.01 (0.6%)	0.02	0.05
15-m (s)	0.96 (0.91–0.98)	1.0 (0.7–1.3)	0.02 (0.7%)	0.03	0.08
20-m (s)	0.97 (0.93–0.98)	1.0 (0.6–1.3)	0.03 (0.8%)	0.04	0.11
25-m (s)	0.97 (0.94–0.99)	0.9 (0.6–1.3)	0.03 (0.8%)	0.04	0.11
30-m (s)	0.98 (0.95–0.99)	0.9 (0.6–1.2)	0.04 (0.9%)	0.05	0.14
*HZT* F_0_ N·kg^−1^	0.69 (0.39–0.86)	3.8 (2.7–4.9)	0.08 (1.2%)	0.30	0.83
V_MAX_ m·s^−1^	0.98 (0.97–0.99)	1.2 (0.8–1.6)	0.11 (1.5%)	0.11	0.30
*HZT* P_0_ W·kg^−1^	0.95 (0.90–0.98)	3.5 (2.5–4.5)	0.32 (2.2%)	0.51	1.41
S_FV_ N·s·m^−1^·kg^−1^	0.84 (0.67–0.93)	4.0 (0.02–0.05)	0.01 (1.1%)	0.04	0.12

*Note.* ICC = intraclass correlation coefficient; CV% = coefficient of variation; SWC = smallest worthwhile change (left side), percentage of mean (right side); SEM = standard error of measurement; MDC_95_ = minimal detectable change at 95% confidence interval.

**Table 5 sports-10-00057-t005:** Mean ± standard deviation (95% confidence interval) and absolute reliability statistics across the average of both days for Musclelab^TM^ Laser Speed; between-day trials.

Performance	Mean ± SD Both Days	ICC	CV%	SWC	SEM	MDC_95_
5-m (s)	1.44 ± 0.04 (1.43–1.47)	0.83 (0.61–0.93)	1.1 (0.5–1.6)	0.01 (0.6%)	0.03	0.08
10-m (s)	2.24 ± 0.07 (2.21–2.27)	0.93 (0.83–0.97)	0.9 (0.6–1.3)	0.01 (0.6%)	0.03	0.08
15-m (s)	2.96 ± 0.11 (2.91–3.01)	0.93 (0.84–0.97)	0.9 (0.5–1.4)	0.02 (0.7%)	0.04	0.11
20-m (s)	3.64 ± 0.15 (3.57–3.70)	0.96 (0.91–0.98)	0.8 (0.5–1.1)	0.03 (0.8%)	0.04	0.11
25-m (s)	4.30 ± 0.19 (4.22–4.38)	0.96 (0.92–0.98)	0.8 (0.5–1.1)	0.03 (0.8%)	0.04	0.11
30-m (s)	4.96 ± 0.23 (4.85–5.06)	0.97 (0.92–0.98)	0.8 (0.5–1.1)	0.04 (0.9%)	0.05	0.14
*HZT* F_0_ N·kg^−1^	7.10 ± 0.36 (6.94–7.26)	0.77 (0.47–0.90)	2.7 (1.8–3.7)	0.07 (1.0%)	0.24	0.66
V_MAX_ m·s^−1^	7.77 ± 0.55 (7.52–8.01)	0.98 (0.95–0.99)	1.1 (0.7–1.5)	0.11 (1.4%)	0.10	0.28
*HZT* P_0_ W·kg^−1^	14.29 ± 1.52 (13.61–14.96)	0.93 (0.83–0.97)	3.1 (2.1–4.1)	0.30 (2.1%)	0.52	1.43
S_FV_ N·s·m^−1^·kg^−1^	−0.87 ± 0.06 (−0.90–−0.84)	0.83 (0.61–0.93)	3.0 (0.01–0.04)	0.01 (1.3%)	0.04	0.11

*Note.* ICC = intraclass correlation coefficient; CV% = coefficient of variation; SWC = smallest worthwhile change (left side), percentage of mean (right side); SEM = standard error of measurement; MDC_95_ = minimal detectable change at 95% confidence interval.

**Table 6 sports-10-00057-t006:** Mean ± standard deviation (95% confidence interval) and absolute reliability statistics for Day 1 and Day 2 for the MySprint app.

Performance	Mean ± SD Day 1	ICC Day 1	CV% Day 1	Mean ± SD Day 2	ICC Day 2	CV% Day 2
5-m (s)	1.44 ± 0.05 (1.41–1.46)	0.80 (0.60–0.91)	2.5 (1.8–3.1)	1.45 ± 0.06 (1.42–1.48)	0.86 (0.72–0.94)	2.2 (1.6–2.8)
10-m (s)	2.21 ± 0.09 (2.17–2.25)	0.94 (0.88–0.97)	1.4 (1.0–1.8)	2.23 ± 0.11 (2.18–2.28)	0.75 (0.59–0.89)	2.2 (0.8–3.6)
15-m (s)	2.91 ± 0.12 (2.85–2.96)	0.96 (0.92–0.98)	1.3 (0.9–1.6)	2.92 ± 0.13 (2.86–2.98)	0.96 (0.92–0.98)	1.2 (0.9–1.6)
20-m (s)	3.58 ± 0.16 (3.51–3.65)	0.95 (0.91–0.98)	1.1 (0.6–1.6)	3.58 ± 0.16 (3.51–3.66)	0.97 (0.95–0.98)	1.1 (0.8–1.3)
25-m (s)	4.21 ± 0.19 (4.12–4.30)	0.98 (0.97–0.99)	0.7 (0.5–1.0)	4.23 ± 0.21 (4.14–4.32)	0.98 (0.96–0.99)	1.0 (0.7–1.2)
30-m (s)	4.85 ± 0.24 (4.74-4.96)	0.98 (0.97–0.99)	0.8 (0.6–1.0)	4.87 ± 0.25 (4.76–4.99)	0.98 (0.96–0.99)	0.9 (0.7–1.2)
*HZT* F_0_ N·kg^−1^	6.89 ± 0.57 (6.63–7.14)	0.83 (0.65–0.92)	5.3 (4.0–6.6)	6.81 ± 0.51 (6.58–7.03)	0.79 (0.58–0.91)	5.0 (3.6–6.3)
V_MAX_ m·s^−1^	8.10 ± 0.61 (7.83–8.37)	0.97 (0.95–0.99)	1.7 (1.2–2.1)	8.06 ± 0.63 (7.78–8.34)	0.98 (0.96–0.99)	1.6 (1.2–2.0)
*HZT* P_0_ W·kg^−1^	14.53 ± 1.79 (13.74–15.33)	0.95 (0.91–0.98)	4.0 (3.1–4.9)	14.32 ± 1.93 (13.46–15.18)	0.95 (0.91–0.98)	4.1 (2.9–5.3)
S_FV_ N·s·m^−1^·kg^−1^	−0.83 ± 0.09 (−0.87–−0.79)	0.79 (0.57–0.90)	7.7 (5.6–9.7)	−0.83 ± 0.07 (−0.86–−0.80)	0.80 (0.59–0.91)	6.1 (4.6–7.5)

**Table 7 sports-10-00057-t007:** Level of agreement of Musclelab^TM^ Laser Speed vs. MySprint app on the sprinting profiling measurements across six trials with 95% confidence interval.

Performance	MLS	MySp	Mean Diff (MLS–MySp)	ICC	*r* _P_	*s*	*i*
5-m (s)	1.45 ± 0.04 (1.43–1.46)	1.44 ± 0.05 (1.42–1.47)	0.001 ± 0.03 (−0.01–0.01)	0.88 (0.72–0.95)	0.81 (0.59–0.92) **	−0.28 (−0.57–0.01)	0.40 (−0.02–0.81)
10-m (s)	2.24 ± 0.07 (2.21–2.27)	2.22 ± 0.09 (2.18–2.26)	0.01 ± 0.05 (−0.003–0.04)	0.90 (0.76–0.96)	0.86 (0.69–0.94) **	−0.2 (−0.51 –0.01)	0.61 (0.05–1.17)
15-m (s)	2.96 ± 0.11 (2.91–3.00)	2.91 ± 0.12 (2.86–2.97)	0.04 ± 0.04 (0.02–0.06) ^¥^	0.93 (0.54–0.98)	0.93 (0.85–0.97) **	−0.11 (−0.27–0.05)	0.36 (−0.12–0.85)
20-m (s)	3.63 ± 0.14 (3.57–3.70)	3.58 ± 0.16 (3.51–3.65)	0.05 ± 0.04 (0.03–0.07) ^¥^	0.95 (0.41–0.98)	0.96 (0.91–0.98) **	−0.68 (−0.19–0.06)	0.30 (−0.16–0.76)
25-m (s)	4.30 ± 0.18 (4.21–4.38)	4.22 ± 0.19 (4.13–4.31)	0.08 ± 0.03 (0.06–0.09) ^¥^	0.95 (−0.01–0.98)	0.98 (0.99–0.95) **	−0.46 (−0.13–0.04)	0.27 (−0.11–0.66)
30-m (s)	4.96 ± 0.23 (4.85–5.06)	4.86 ± 0.24 (4.75–4.97)	0.09 ± 0.04 (0.07–0.11) ^¥^	0.95 (−0.01–0.99)	0.98 (0.97–0.95) **	−0.05 (−0.12–0.02)	0.36 (−0.003–0.72)
*HZT* F_0_ N·kg^−1^	7.10 ± 0.36 (6.94–7.26)	6.85 ± 0.50 (6.62–7.07)	0.25 ± 0.46 (0.04–0.45) ^¥^	0.55 (−0.002–0.81)	0.44 (0.04–0.73) *	−0.44 (–0.99–0.10)	3.37 (−0.45–7.2)
V_MAX_ m·s^−1^	7.77 ± 0.55 (7.52–8.01)	8.08 ± 0.61 (7.81–8.35)	−0.31 ± 0.13 (−0.37–−0.25) ^¥^	0.92 (−0.08–0.84)	0.98 (0.95–0.99) **	−0.11 (−0.20–−0.02)	0.55 (−0.15–1.29)
*HZT* P_0_ W·kg^−1^	14.29 ± 1.52 (13.61–14.96)	14.43 ± 1.81 (13.62–15.23)	−0.14 ± 0.77 (−0.48–0.20)	0.94 (0.86–0.97)	0.90 (0.78–0.96) **	−0.18 (−0.38–0.02)	2.45 (−0.52–5.4)
S_FV_ N·s·m^−1^·kg^−1^	−0.87 ± 0.06 (−0.89–−0.84)	−0.83 ± 0.07 (−0.87–−0.80)	−0.04 ± 0.06 (−0.06–−0.01) ^¥^	0.68 (0.22–0.86)	0.60 (0.24–0.81) **	−0.34 (−0.79–0.11)	−0.32 (−0.71–0.06)

*Note.* ICC = intraclass correlation coefficient; *r*_P_ = Pearson product moment correlation coefficient; *s* = slope of the regression line; *i* = intercept of the regression line. ^¥^
*p* < 0.05 (2-tailed), * *p* < 0.05 (1-tailed), ** *p* < 0.01 (1-tailed).

## Data Availability

The date presented in this study are available upon request from the corresponding author.

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
