# Peer review of "Level of Agreement, Reliability, and Minimal Detectable Change of the MusclelabTM Laser Speed Device on Force–Velocity–Power Sprint Profiles in Division II Collegiate Athletes"

_sports, 2022, doi:10.3390/sports10040057_

Round 1

Reviewer 1 Report

This work examines the validity, reliability and accuracy of a laser-based commercial device to measure running speed (MuscleLab) and compares it with a phone-camera-based app (MySp), of college 22 athletes of 3 different sports. They found high agreement with measurements between and within days and low correlation between the methods in one estimated variable (horizontal power). There are a few comments below that require the authors' attention and regard the quality and importance of their work.

The introduction is quite general and should focus more on the topic of the study, which is the first line of the title. The authors should comment on the previous related work of van den Tilaar (https://pubmed.ncbi.nlm.nih.gov/34640882/, https://pubmed.ncbi.nlm.nih.gov/34640882/) which has a very similar topic, evaluating the same device.
Although, one of the above works are mentioned in the introduction, the authors should clearly describe the findings of these papers, and make clear what is new the current study provides. 
The authors should better justify why their work is original and in which sense their findings are important. In the current status, this work lacks in originality.

Another issue is the usage of the MySp method as reference for the evaluation of sprint. It is not clear if this would be the best method to use as "gold standard". There might be a random bias due to the pole adjustments (L177-8). Possibly this is the reason for the observed low correlation between the methods. The selection of this method as reference is stated as limitation with the argument that it was easily accessible, but still there are better methods to validate a device, and this was the main purpose of this work.

I am missing the accuracy of MySp, which has been used as reference. At least in the discussion, this is not commented.

Furthermore, it should be stated if all measurements for all athletes were assessed during the same day. 

The results should have units after each number.

Authors of other studies should be referred with their name (not numbers)

Minor issues

eq.1: What does Vh stand for? Please be consistent with notations (see eq.3 F(h))

Please revise for language (e.g. L162, L147 etc)

Figure 2: what do the yellow lines represent?

Reviewer 2 Report

GENERAL COMMENTS

Thank you for the opportunity to review your work. The manuscript evaluate the level of agreement, reliability, and minimal detectable change of the MusclelabTM Laser Speed (MLS) on Force-Velocity-Power profile (FVP-profile) in 30-meter sprint. The level of agreement is calculated by comparing it to the MySprint app. Given the widespread use among physical trainers and athletes of the FVP-profile, I do believe there is merit in this area of research. However, I find some methodological issues that should be carefully addressed before I can recommend it for publication or not.

  • I think the study is incomplete and it could be completed with more possible calculations. I suggest that authors consider calculating the slope of the PV-profile (Sfv). For a complete interpretation of FVP-profile, Sfv is necessary. In fact, this parameter is used to determine whether the profile is optimal or not. The authors justify their non-calculation because the MLS software does not return it by default. But its calculation is possible from HZT F0 and Vmax. I think it's interesting to know how the low level of HZT F0 agreement and Vmax bias between MLS and Mysprint affects to Sfv calculation.
  • The authors calculate the level of agreement between the MLS and Mysprint. The authors establish that Mysprint is a valid tool. But the authors do not use the same methodology to calculate the FVP-profile that appears in the references which validate Mysprint (Romero-Franco et al. 2017). The main drawbacks are focussed on the lower sampling rate used (120 fps vs 240 fps) and the type of sprint start (two-point vs three-point). These modifications could compromise the validity of Mysprint. The authors should establish to what extent these methodological modifications affect their results.
  • I think that the type of sprint start used and the way to determine the sprint start could invalidate the data, especially HZT F0 data. From the literature (Morin and Samozino, 2018) it is suggested the three-point crouching start and add 0.1 s to each split time measures as the best method. I wonder if those 0.1 s are also valid for two-point start. Previous studies have added to all sprint times 0.5 s was added to all sprint times when a two-point sprint start was used (Haugen et al. 2020). I suggest that authors justify whether those 0.1 s that Mysprint automatically adds in the calculations of split times are also valid for a two-points sprint start.
  • From the literature it is suggested that only data from start to Vmax be used to calculate the FVP-profile when using instantaneous velocity systems (Morin and Samozino, 2018). The rest of the data must be deleted to avoid errors. In addition, because the determination of the start is complicated by signal noise, it is recommended to delete all the values for which there is a doubt, and add a time delay to the mathematical function which calculate Vh(t). The manuscript does not indicate whether these recommendations have been carried out. In the case that these recommendations cannot be applied in the MLS software, I suggest that authors consider exporting the raw data and performing the calculations externally. This way you can know if the levels of agreement and bias detected are due to MLS software or data capture.

Round 2

Reviewer 1 Report

After reading the revised version I can confirm that the authors took into consideration the reviewer's comments and made the appropriate changes. The scope of the study and the added value is clearer.

There are only some minor issues that I would like to point out.

- I agree with adding in the revised version the slope of the force-velocity curve (SFV). However, what is missing is the figure with the Bland-Altman graph regarding this parameter.

- To increase readability in tables 5 and 6, the 95% confidence intervals should be in a second line.

- Please use a common definition for the app (MySp or MySprint) in the whole manuscript.

- Please add in the caption description of figure 2 that the yellow lines represent the marking poles.

Reviewer 2 Report

Thank you for allowing me to review the revised manuscript and for taking the time to respond to comments. The authors have made great improvements on the manuscript, but continue to exist some limitations of the study which must be addressed before recommending its publication. Specifically the sample rate and the start set position used. I suggest to the authors that both limitations be addressed in more detail in the discussion section.

I have no doubt that a 120 Hz camera is not considered a high-speed camera. But if MySprint validation studies have used with a 240 Hz camera and the authors use a camera with half the sampling rate, their results could be conditioned by this circumstance. In fact, I think it might be interesting to know what the minimum sample rate is for the results to be valid. Brain food…

Similarly, if we consider:

- That in the literature the main source of error is the way to determine the sprint start.

- That this must be corrected not only based on the type of start position used, but also, based on the technology used to collect the data.

- That the lowest accuracy reported are those corresponding to the HZT F0, which is calculated based on the data recorded at the beginning of the sprint.

I think these are important aspects that need to be addressed with more detail than the current one. 
